# Developing a Mathematical Model for Nucleate Boiling Regime at High Heat Flux

**Mohd Danish \* and Mohammed K. Al Mesfer**

Chemical Engineering Department, College of Engineering, King Khalid University, Abha 61411, Saudi Arabia; almesfer@kku.edu.sa
\* Correspondence: danishmohd111@yahoo.com

**Abstract:** A mathematical model has been developed for heat exchange in nucleate boiling at high flux applying an energy balance on a macrolayer. The wall superheat, macrolayer thickness, and time are the parameters considered for predicting the heat flux. The influence of the wall superheat and macrolayer thickness on average heat flux has been predicted. The outcomes of the current model have been compared with Bhat's constant macrolayer model, and it was found that these models are in close agreement corresponding to the nucleate pool boiling regime. It was concluded that the wall superheat and macrolayer thickness contributed significantly to conduction heat transfer. The average conduction heat fluxes predicted by the current model and by Bhat's model are in close agreement for a thinner macrolayer of approximately 50 μm. For higher values of the wall superheat, which corresponds to the nucleate pool boiling condition, the predicted results strongly agree with the results of Bhat's model. The findings also validate the claim that conduction across the macrolayer accounts for the main heat transfer mode from the heater surface to boiling liquid at high heat flux in nucleate pool boiling.

**Keywords:** mathematical model; energy balance; macrolayer; heat flux

---

## 1. Introduction

Nucleate pool boiling is a significant boiling regime since it contributes considerably to a high heat transfer rate. The nucleate boiling's mechanism is vastly complex due to its dependence on numerous parameters. The region of isolated bubbles and the region of interference are two boiling regions identified by the investigator [1] relying on the experimental study. An illustrative study at atmospheric pressure of pool boiling was also conducted [2] and it was suggested that the vapor structure on the heater surface advanced through a succession of discrete bubbles, then vapor columns, and eventually vapor patches with an increased surface temperature. It was also deduced that any heat transfer model relying on the dynamics of individual bubbles should be in significant error. The investigators [3] hypothesized about the rough differentiation of different regions of boiling and the macrolayer formation mechanism. Good agreement was found between the theoretical initial macrolayer thickness and the experimental literature values. The researchers [4] previously achieved findings comparable to those described by the investigators [2]. The numbers of bubble-generating sites were estimated all over the nucleate region and it was suggested that the heat flux is proportional roughly to the square root of the number of active sites. A graphic investigation at atmospheric pressure showed that a minimum of three and probably four regions exist in nucleate boiling, relying on the vapor generation mode. A mechanism [5] for nucleate pool boiling was suggested and it was shown that a liquid film exists on the heated surface. It was concluded that a pool boiling crisis is associated to a sporadic response of vapor masses and to the depletion of liquid film that exists on a heated surface.

Moissis and Berenson [6] carried out a study on the hydrodynamic transitions in nucleate boiling. The first transition is ruled by a continuity equation and the second transition governs the maximal heat flux. Dhir [7] proposed a governing mechanism of nucleate boiling and also concluded that the absolute numerical solution of bubble dynamics and the related heat transfer processes seem to represent a vigorous perspective. The importance of nucleate pool boiling arises from its capability to accomplish high heat transfer at a comparatively low thermal driving force. Enhancing the heat fluxes in nucleate boiling was studied exclusively [8,9]. An analytical mechanism of enhancing the boiling heat transfer by assimilating low-conductivity materials at the surface-liquid was reported. It was found that replacing 18% of the surface with a non-conductive epoxy leads to more than 5× increase in the heat exchange rate [8]. A combined thermal-hydraulic model [9] was developed relating the critical heat flux (CHF) to the rewetting of a hot dry spot on a boiling surface, thereby divulging the mechanism ruling the CHF enhancement.

A stated supposition is essential to model the boiling process. Vapor-liquid exchange models, macrolayer/microlayer and latent heat evaporation models, and bubble agitation models are models classified relying on the supposition of the heat transfer. The bubble agitation model is relying on the assumption that bubble growth and release cause the closer-by liquid to experience the heat exchange process [10,11]. The predictions of the models relying on the vapor-liquid exchange notion differ significantly from tangible results [12,13]. The use of nanofluid as an alternative of base fluid enhances the heat transfer coefficient applying the Lattice Boltzmann method [14,15]. The computational fluid dynamics (CFD) approach was used to forecast the heat transfer coefficient at a high heat flux [16]. The findings of numerical experiments were compared with calculated values applying empirical correlations and it was suggested that the heat transfer coefficient robustly depends on wall roughness.

Son and colleagues [17] suggested that surface roughness contributed significantly to heat transfer using a numerical simulation. A 3-D numerical study [18] of saturated water was performed under atmospheric conditions and it was revealed that a vapor blanket forms on the top heater surface under a high heat flux condition that leads to a temperature rise of the heater surface. The boiling curve is obtained numerically by determining the macrolayer thickness relying on the macrolayer model [19]. It is also highlighted that a macrolayer plays a significant role in enhancing the transient CHF. The nucleate boiling was investigated by utilizing the 2-D model based on the height function model and volume of fluid (VOF) [20]. The growth of the bubble size and localized heat transfer were measured relying on numerical methods. It was also concluded that vapor phase in the proximity of the heating wall is superheated. Within the frame of CFD, a model based on a microlayer was proposed to simulate nucleate boiling and the suggested model has been ratified for pool boiling of water at atmospheric pressure [21]. Furthermore, it was found that the temperature distribution over the heat transfer surface agrees very well with experimental data.

The simulated results agree very well with experimental findings in a CFD study of cryogenic fluid by Xiaobin and others [22]. Further, the effects of the wall superheat on the bubble diameter were precisely investigated along with the determination of the heat flux and superheat. The correlation was proposed to foretell the departure diameter. A numerical technique was used to examine the nucleate heat transfer on a lean substrate with a horizontal test submersed in a saturated liquid having artificial cavities [23]. The hybrid code utilized in the investigation allows enormous simulations in a rational computational time. The quantitative effects of the parameters on the detachment characteristics using a better VOF approach have been studied and it was verified that the initial thermal boundary layer is a consequential parameter in the growth of the bubble and detachment process [24]. The characteristics of bubble detachment appear to be notably influenced by the triple-line contact angle for all working fluids. The heat exchange through the boundary between the vapor, liquid, and solid phases was investigated by a new developed model [25]. It was concluded that the predicted results agree very well with experimental data for the partial nucleate boiling regime for different fluids and the model can be contemplated for fully developed nucleate boiling. The ethanol-water system was used to identify the microlayer contribution on a heat exchange using a numerical simulation [26]. The laser

extinction method was applied to estimate the distribution of the macrolayer formed under a growing bubble using a specifically designed system.

The complete range of nucleate boiling was studied by proposing a generalized empirical correlation [27]. The coefficient value utilized in the correlation was determined by the least square technique and the non-linear solution technique was used to develop the correlation. The heat transfer coefficient was predicted using $TiO_2$-water nanofluids at different low concentrations [28]. The aluminum and copper circular plates with varying surface roughness were utilized as heating surfaces and it was suggested that the developed correlation can be used for prediction at a certain level. Chu and colleagues [29] developed a comprehensive model for the fractal distribution of nucleation sites existing on heating surfaces and the proposed model was demonstrated as a function of sizes of active nucleation sites, superheat, and fluid properties. Vapor bubble growing on a heated wall was studied using static and dynamic contact models [30]. The vapor-liquid interface was apprehended with the help of a level-set technique and it was revealed that the contact angle at the base of the bubble differs during the ebullition cycle. A theoretical model very well described the hydrodynamic behavior of the bubble's vapor-liquid interface at the heater surface [31]. The momentum flux was recognized as a notable parameter and CHF of the refrigerants, cryogenic liquid and water were very well predicted by the proposed model.

A transient model for heat exchange based on a macrolayer was suggested by researchers [32,33]. The maximal heat flux enhances and the nucleate boiling region is increased along with the steady state boiling curve on raising the heating speed [33]. The notion that the macrolayer model is mainly suitable for the high heat flux regime was also proposed by He and others [19]. The evaporation of the macro-microlayer plays an imperative role, particularly at the high flux. An alternate critical heat flux (CHF) theory that relied on the macrolayer was suggested by [34,35]. In the study [34], the Helmholtz instability was applied on the vapor stems' vapor-liquid interface distributed in a liquid layer. The investigation of CHF was illustrated for both pool boiling and forced convection, suggesting the justifiability of the physical structure of model. The model [35] was sufficiently able to predict the boiling curve transition. The prediction of the time fraction and liquid contact concurs fairly well with experimental data. It was also suggested that surface coating enhanced the transition boiling heat transfer and raised the wall superheat. These proposed models hold the primary elements of Zuber's model [36]. In Zuber's model [36], the transition boiling was investigated by examining the stability of the plane vortex separating two inviscid fluids. The expressions were derived to predict the maximal and minimal heat exchange rates for nucleate and film boiling regimes. A theoretical model was suggested by Bhat and others [3] and recommended that active site density may play a significant role in CHF and nucleate boiling. Dhir and Liaw [37] proposed that wall heat is conducted into a liquid macrolayer near the stems using the time-averaged model. The 2-D conduction was used to estimate the heat transfer rate into the thermal layer and temperature distribution inside the layer. It was also suggested that maximal heat flux predicted by the model are acceptable for the surfaces which are properly wetted. Bhat et al. [38,39] developed an analytical heat transfer model, assuming the constant macrolayer thickness and obtained the following expressions: Equations (1)–(3) of the temperature profile, conduction heat flux, and average conduction heat flux, respectively.

$$\theta = L^{-1}(\bar{\theta}) = \sum_{n=0}^{\infty} \text{erfc}\left[\frac{\delta(2n+1) - y}{2(\alpha t)^{1/2}}\right] - \sum_{n=0}^{\infty} \text{erfc}\left[\frac{\delta(2n+1) + y}{2(\alpha t)^{1/2}}\right] \tag{1}$$

$$q_c = \frac{k\Delta T}{(\pi\alpha)^{1/2}}\left\{\frac{1}{t^{1/2}} + 2\sum_{n=1}^{\infty}\left(\frac{e^{-(n\delta\beta)^2/t}}{t^{1/2}}\right)\right\}, \beta = \alpha^{1/2} \tag{2}$$

$$q_{avg} = \frac{4k\Delta T}{t_c(\pi\alpha)^{1/2}}\left\{\frac{(t_c)^{1/2}}{2} + \sum_{n=1}^{\infty} t_c^{1/2}e^{-c/t_c} - (\pi c)^{1/2}\text{erf}(c/t_c)^{1/2}\right\} \tag{3}$$

The heat exchange through the macrolayer by maintaining the heater bottom at a constant input heat was studied by Prasad [40]. Jairajpuri and Saini [41] proposed a 1-D analytical model of reducing the macrolayer thickness. The methanol-water system was used to predict the conduct heat flux and it was suggested that a dimensionless parameter $Z_O$ may be used as an indicator of the combined result of the physical parameters initial wall superheat, initial macrolayer thickness, and wall heat flux on the conduction heat flux. Prasad and colleagues [42] proposed an analytical model considering the heat exchange in both the macrolayer and heater wall. It was concluded that the heat transfer rate can decrease or increase with increased heater wall thickness relying on the non-dimensional parameter. A detailed in-depth study was conducted, taking into consideration the heat exchange and bubble dynamics [43].

The significance of the hydrophilic/hydrophobic surface was analyzed by Mesoscale simulations of boiling curves [44]. It was observed that boiling curves are similar in nucleate and film boiling regimes under constant wall heat flux/constant wall temperature conditions. The saturated temperature has comparatively an insignificant effect on the boiling curve upto the regime of fully developed boiling. A numerical simulation was performed by incorporating a moderate Jakob number, contact angle, and a high-density ratio between phases [45]. The effect on the numerical solution of the thermal heat exchange in a solid heater was also evaluated. It was reported that there is no influence in the design of the thick and most conductive material. A numerical simulation of the nucleate boiling using $H_2O$ and $H_2O$-silica nanofluid was proposed [46] with the help of Eulerian multiphase approach, and the numerical results were compared with the experimental results to validate the suggested approach. The bubble diameter was predicted by the classical heat flux partitioning model and it was also suggested that nanoparticle depositions on the heater surface leads to increased performance of heat transfer in nanofluid. An analytical solution of the model was presented for heat transfer through a macrolayer at a high heat flux, and the predicted results were compared with the experimental results [47].

The effect of microgravity conditions on boiling heat exchange by Mesoscale simulations has been investigated [48]. The bubble departure diameter and the period of one bubble cycle increased with a reduced gravity level and it was also concluded that gravity has a notable influence on pool boiling curves. A correlation that incorporates the influence of the surface/liquid combination to predict the heat transfer coefficient at a high flux was suggested [49]. The proposed correlation provides a good prediction of the boiling behaviour of fluid with considerably different thermophysical properties under confined and unconfined conditions. An algorithm based on the VOF method evolved [50] to study single bubble behavior and microlayer evaporation. It was proposed that a high thermal conductivity plate leads to the effective heat supply to the heat transfer surface and nearly 30% of the bubble volume is occupied by microlayer evaporation. An analytical study [51] of heat transfer in the nucleate regime of pool boiling using R124a/R245fa zeotropic mixture was carried out. The three-stage model of nucleate boiling was recommended and it was found that zeotropic mixture contributes to higher CHF compared with a pure component. The objective of present study is to develop a mathematical model at high heat flux under a nucleate boiling condition using the energy balance. The dependence of the wall superheat, macrolayer thickness, and time on heat flux is analyzed. The contribution of the wall superheat and macrolayer thickness to the conduction heat flux is analyzed. The effect of the wall superheat and macrolayer thickness on the average conduction heat flux is also investigated under constant conditions of the total departure time. The findings of the present model are compared with the existing model developed by Bhat et al. [38].

## 2. Model Developments

Energy balance:

$$[\text{Energy transported in through bulk flow}] + [\text{Energy transported in through thermal diffusion}] + [\text{Net generation or loss}] = [\text{Energy transported out by bulk flow}] + [\text{Energy out by thermal diffusion}] + [\text{Net accumulation}] \tag{4}$$

Energy transported in through bulk flow:

$$= \left.\left(\rho u\, \Delta y\, \Delta z\, \Delta t\, c_p \Delta T\right)\right|_x + \left.\left(\rho v\, \Delta x\, \Delta z\, \Delta t\, c_p \Delta T\right)\right|_y + \left.\left(\rho w\, \Delta x\, \Delta y\, \Delta t\, c_p \Delta T\right)\right|_z \tag{5}$$

Energy transported in through thermal diffusion:

$$= \left.\left(-k\, \Delta y\, \Delta z\, \Delta t\, \frac{\partial T}{\partial x}\right)\right|_x + \left.\left(-k\, \Delta x\, \Delta z\, \Delta t\, \frac{\partial T}{\partial y}\right)\right|_y + \left.\left(-k\, \Delta x\, \Delta y\, \Delta t\, \frac{\partial T}{\partial z}\right)\right|_z \tag{6}$$

$$\text{Let net generation or loss} = S_r \tag{7}$$

Similarly, Energy out by bulk flow:

$$= \left.\left(\rho u\, \Delta y\, \Delta z\, \Delta t\, c_p \Delta T\right)\right|_{x+\Delta x} + \left.\left(\rho v\, \Delta x\, \Delta z\, \Delta t\, c_p \Delta T\right)\right|_{y+\Delta y} + \left.\left(\rho w\, \Delta x\, \Delta y\, \Delta t\, c_p \Delta T\right)\right|_{z+\Delta z} \tag{8}$$

Energy out by thermal diffusion:

$$= \left.\left(-k\, \Delta y\, \Delta z\, \Delta t\, \frac{\partial T}{\partial x}\right)\right|_{x+\Delta x} + \left.\left(-k\, \Delta x\, \Delta z\, \Delta t\, \frac{\partial T}{\partial y}\right)\right|_{y+\Delta y} + \left.\left(+ \Delta x\, \Delta y\, \Delta t\, \frac{\partial T}{\partial z}\right)\right|_{z+\Delta z} \tag{9}$$

Net accumulation of heat:

$$= \left.\left(\rho c_p \Delta x\, \Delta y\, \Delta z\ \Delta T\right)\right|_{t+\Delta t} + \left.\left(\rho c_p \Delta x\, \Delta y\, \Delta z\ \Delta T\right)\right|_t \tag{10}$$

Putting the values of Equation (5) through Equation (10) into Equation (4), we get

$$
\begin{array}{c}
\left.= \left(\rho u\, \Delta y\, \Delta z\, \Delta t\, c_p \Delta T\right)\right|_x \quad + \quad \left.\left(\rho v\, \Delta x\, \Delta z\, \Delta t\, c_p \Delta T\right)\right|_y \quad + \quad \left.\left(\rho w\, \Delta x\, \Delta y\, \Delta t\, c_p \Delta T\right)\right|_z \\[4pt]
\hline
\left.= \left(-k\, \Delta y\, \Delta z\, \Delta t\, \tfrac{\partial T}{\partial x}\right)\right|_x \quad + \left.\left(-k\, \Delta x\, \Delta z\, \Delta t\, \tfrac{\partial T}{\partial y}\right)\right|_y \quad + \left.\left(-k\, \Delta x\, \Delta y\, \Delta t\, \tfrac{\partial T}{\partial z}\right)\right|_z \quad + S_r \\[4pt]
\hline
\left.= \left(\rho u\, \Delta y\, \Delta z\, \Delta t\, c_p \Delta T\right)\right|_{x+\Delta x} + \left.\left(\rho v\, \Delta x\, \Delta z\, \Delta t\, c_p \Delta T\right)\right|_{y+\Delta y} + \left.\left(\rho w\, \Delta x\, \Delta y\, \Delta t\, c_p \Delta T\right)\right|_{z+\Delta z} \\[4pt]
\left.= \left(-k\, \Delta y\, \Delta z\, \Delta t\, \tfrac{\partial T}{\partial x}\right)\right|_{x+\Delta x} + \left.\left(-k\, \Delta x\, \Delta z\, \Delta t\, \tfrac{\partial T}{\partial y}\right)\right|_{y+\Delta y} + \left.\left(-k\, \Delta x\, \Delta y\, \Delta t\, \tfrac{\partial T}{\partial z}\right)\right|_{z+\Delta z} \\[4pt]
\hline
\left.= \left(\rho c_p \Delta x\, \Delta y\, \Delta z\ \Delta T\right)\right|_{t+\Delta t} \qquad\qquad \left.+ \left(\rho c_p \Delta x\, \Delta y\, \Delta z\ \Delta T\right)\right|_{t}
\end{array}
\tag{11}
$$

Dividing Equation (11) by $\Delta x\, \Delta y\, \Delta z$ and taking the limit as $\Delta t \to 0$:

$$
\begin{aligned}
&\frac{\partial}{\partial x}\left(\rho u\, c_p\, T\right) + \frac{\partial}{\partial y}\left(\rho v\, c_p\, T\right) + \frac{\partial}{\partial z}\left(\rho w\, c_p\, T\right) + \frac{\partial}{\partial t}\left(\rho c_p\, T\right) \\
&= \frac{\partial}{\partial x}\left(k\, \frac{\partial T}{\partial x}\right) + \frac{\partial}{\partial y}\left(k\, \frac{\partial T}{\partial y}\right) + \frac{\partial}{\partial z}\left(k\, \frac{\partial T}{\partial z}\right) + S_r
\end{aligned}
\tag{12}
$$

The above equation can also be written in vector notation as follows:

$$
\left(\nabla.\rho u\, C_p\, T\right) + \frac{\partial}{\partial t}\left(\rho\, C_p\, T\right) = \nabla^2 .(kT) + S_r
\tag{13}
$$

The density $(\rho)$, specific heat $(C_p)$, and thermal conductivity (k) are assumed to remain constant, and hence, the above equation reduces to the following form:

$$
\rho\, c_p \frac{\partial T}{\partial t} + \rho c_p\, u\, \frac{\partial T}{\partial x} + \rho c_p\, v\, \frac{\partial}{\partial v} + \rho c_p\, w\, \frac{\partial T}{\partial w} = k\frac{\partial^2 T}{\partial x^2} + k\frac{\partial^2 T}{\partial y^2} + k\frac{\partial^2 T}{\partial z^2} + S_r
\tag{14}
$$

Assuming that there is no bulk transport in the x and z direction i.e., $u \to 0$, $w \to 0$, hence, the above equation is of the following form:

$$
\rho\, c_p \frac{\partial T}{\partial t} + \rho\, c_p\, \overset{0}{\cancel{u\, \frac{\partial T}{\partial x}}} + \rho\, c_p\, v\, \frac{\partial T}{\partial y} + \rho\, c_p\, \overset{0}{\cancel{w\, \frac{\partial T}{\partial w}}} = \rho\, c_p\, k\frac{\partial^2 T}{\partial x^2} + k\frac{\partial^2 T}{\partial y^2} + k\frac{\partial^2 T}{\partial z^2} + S_r
$$

i.e.,

$$
\rho\, c_p \frac{\partial T}{\partial t} + \rho\, c_p\, v\, \frac{\partial T}{\partial y} = k\frac{\partial^2 T}{\partial x^2} + k\frac{\partial^2 T}{\partial y^2} + k\frac{\partial^2 T}{\partial z^2} + S_r
\tag{15}
$$

Assuming that there is no variation in the temperature in the x and z direction and there is no net heat generation or loss in the macrolayer due to no heat source $(S_r = 0)$, the following simplified expression is obtained:

$$\rho \, c_p \frac{\partial T}{\partial t} + \rho \, c_p \, v \, \frac{\partial T}{\partial y} = k\frac{\partial^2 T}{\partial x^2} + k\frac{\partial^2 T}{\partial y^2} + k\frac{\partial^2 T}{\partial z^2}$$
$$\text{i.e., } \rho \, c_p \frac{\partial T}{\partial t} + \rho \, c_p \, v \, \frac{\partial T}{\partial y} = k\frac{\partial^2 T}{\partial y^2} \tag{16}$$

Dividing (16) by $\rho \, c_p$, the above equation is obtained as:

$$\frac{\rho c_p}{\rho c_\rho}\frac{\partial T}{\partial t} + \frac{\rho c_p}{\rho c_\rho}v \frac{\partial T}{\partial t} = \frac{k}{\rho c_\rho}\frac{\partial^2 T}{\partial y^2} \Longrightarrow \frac{\partial T}{\partial t} + v\frac{\partial T}{\partial y} = \frac{k}{\rho c_\rho}\frac{\partial^2 T}{\partial y^2} \tag{17}$$

Put $\frac{k}{\rho c_\rho} = \alpha = $ thermal diffusivity, above equation reduces as follows:

$$\Longrightarrow \frac{\partial T}{\partial t} + v\frac{\partial T}{\partial y} = \propto \frac{\partial^2 T}{\partial y^2}$$
$$\Longrightarrow \alpha\frac{\partial^2 T}{\partial y^2} - v\frac{\partial T}{\partial y} = \frac{\partial T}{\partial t} \tag{18}$$

—transient one dimensional heat conduction equation.

The liquid-vapor interface conduction heat flux, responsible for evaporation:

$$q_c = -k\left(\frac{\partial T}{\partial y}\right)_{y=\delta} \tag{19}$$

## 3. Analytical Solution

1-D transient heat conduction Equation (18) in dimensional form:

$$\alpha \, \frac{\partial^2 \theta}{\partial y^2} - v \, \frac{\partial \theta}{\partial y} = \frac{\partial \theta}{\partial t} \, ; \, 0 \leq y \leq \delta \, ; \, t > 0 \tag{20}$$

I.Cs,

$$\theta(y, \, 0) = 0 \, ; \, 0 \leq y \leq \delta \tag{21}$$

Spatial B.Cs as,

$$\theta(0, \, t) = 0 \, ; \, t > 0 \tag{22}$$

$$\theta(\delta, \, t) = 1 \, ; \, t > 0 \tag{23}$$

And

$$\theta = \frac{T_w - T}{T_w - T_s}; \tag{24}$$

is liquid vapour interface velocity (25)

$$v = \frac{q_w}{\rho_l h_{fg}} \tag{25}$$

Equation (20) was solved analytically using I.Cs and B.Cs with the help of Equation (25) and the following solution was obtained as presented in Equation (26)

$$\theta(y,t) = e^{\frac{v}{2\alpha}(y-\delta)}\left[\frac{\sinh \frac{v}{2\alpha}y}{\sinh \frac{v}{2\alpha}\delta} + \sum_{n=1}^{\infty}\frac{2n\pi(-1)^n \sin n\pi y/\delta}{\left(\frac{v^2}{4\alpha} + \frac{n^2\pi^2\alpha}{\delta^2}\right)\exp\left(\frac{v^2}{4\alpha} + \frac{n^2\pi^2\alpha}{\delta^2}\right)t}\right] \tag{26}$$

Using Equation (19), the expression for the conduction heat flux reduces to:

$$q_c = k\Delta T \left[ \frac{v}{\alpha\left(1 - \exp\left(\frac{-v\delta}{\alpha}\right)\right)} + \sum_{n=1}^{\infty} \frac{2n^2\pi^2}{\delta\left(\frac{v^2\delta^2}{4\alpha^2} + n^2\pi^2\right)\exp\left(\frac{v^2}{4\alpha} + \frac{n^2\pi^2\alpha}{\delta^2}\right)t} \right] \qquad (27)$$

Let $t_c$ = total cycle time, the average conduction heat flux is calculated as:

$$q_{avg} = \frac{1}{t_c}\int_0^{t_c} q_c(t)dt \qquad (28)$$

Average heat flux is obtained is given as:

$$q_{avg} = \frac{k\Delta T}{t_c} \left\{ \frac{vt_c}{\alpha\left(1 - \exp\left(\frac{-vs}{\alpha}\right)\right)} - \sum_{n=1}^{\infty} \frac{2n^2\pi^2\alpha\left\{\exp -\left(\frac{v^2}{4\alpha} + \frac{n^2\pi^2\alpha}{\delta^2}\right)t - 1\right\}}{\delta^3\left(\frac{v^2}{4\alpha} + \frac{n^2\pi^2\alpha}{\delta^2}\right)^2} \right\} \qquad (29)$$

## 4. Results and Discussion

### 4.1. Instantaneous Heat Transfer Rates

Figure 1 [38] depicts the dependence of the heat flux on the wall superheat at different conditions. The data were collected for a departure time of 5 ms at constant macrolayer thicknesses of 50 μm, 150 μm, and 200 μm. All these curves demonstrate that regardless of the parameter values, the heat flux strongly depends on the wall superheat. The maximal conduction heat flux of approximately 1.3601 MW/m$^2$ was obtained for the macrolayer thickness of 50 μm. An almost constant heat flux of 0.6623 MW/m$^2$ was noticed for the thicknesses of 150 μm and 200 μm, indicating the insignificance of a thicker macrolayer for the heat transfer. The thicker macrolayers contribute practically the same heat conduction fluxes as depicted for macrolayer thicknesses of 150 μm and 200 μm. It is also obvious that thinner macrolayers have consistently higher heat exchange rates.

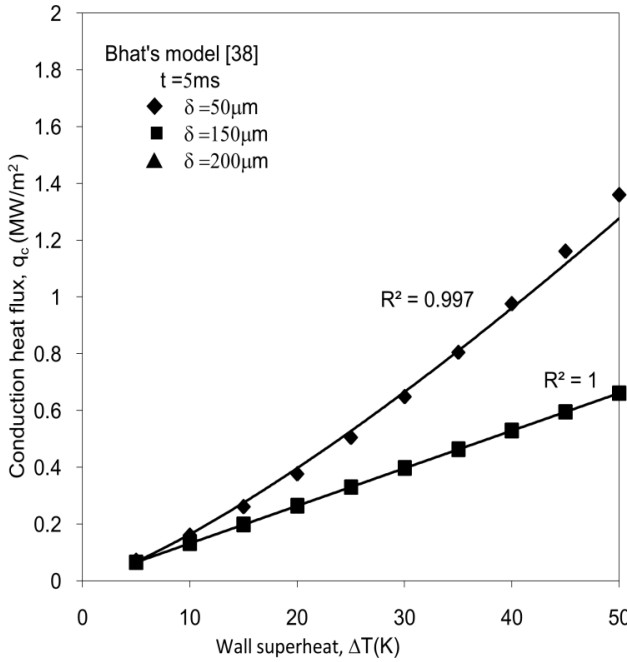

**Figure 1.** Effect of the wall superheat on conduction heat flux for different macrolayer thicknesses.

Figure 2 (present model) shows the dependence of the instantaneous conduction heat transfer rate on the wall superheat. The data were obtained for a departure time of approximately 1 ms with a macrolayer thickness corresponding to 50 µm, 150 µm, and 200 µm. The curves depicted in Figure 2 show that the heat transfer rate is strongly relying on the wall superheat. An increased wall superheat leads to higher rates of heat transfer. The maximal and minimal conduction heat fluxes of 1.5167 MW/m$^2$ and 1.3535 MW/m$^2$ were obtained for macrolayer thicknesses corresponding to 50 µm and 200 µm, respectively, at 50 K superheat. It is also clearly demonstrated that at the same superheat, the thinner macrolayer contributes to increased heat energy. The effect of the macrolayer thickness on the wall superheat is less pronounced in the present model when compared with Bhat's constant macrolayer model.

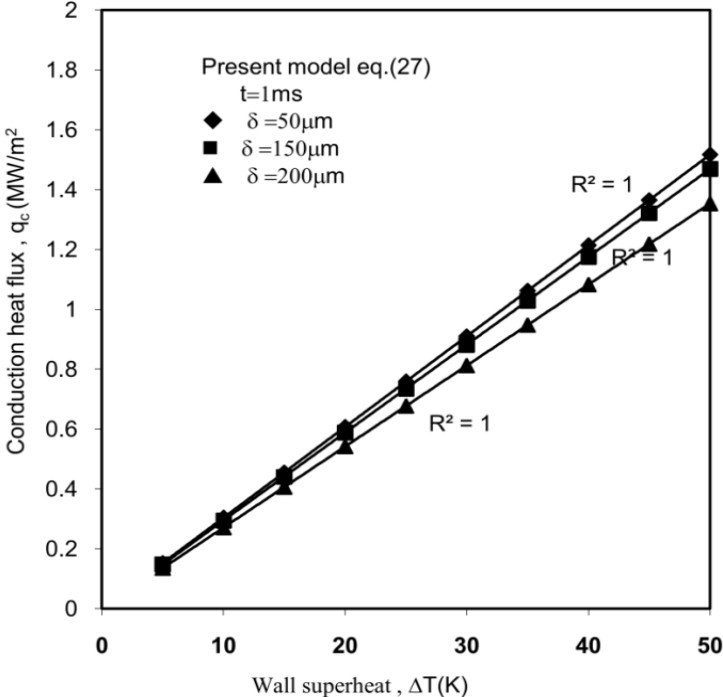

**Figure 2.** Effect of the wall superheat on the conduction heat flux for different macrolayer thicknesses.

Figure 3 [38] depicts the effect of the macrolayer on the conduction heat transfer rate. The data were obtained for the wall superheats of 20 K, 30 K, and 40 K with a time corresponding to 10 ms. The heat exchange rate reduces with enlarged macrolayer thickness. The thicker macrolayer transports less heat energy as compared with the thinner macrolayer. The conduction heat flux decreases from 2.0942 MW/m$^2$ to 0.2094 MW/m$^2$ with increased macrolayer thickness from 30 µm to 200 µm, respectively, under a constant condition of 40 K superheat. The dependence of the macrolayer thickness on the conduction flux is less evident for smaller wall superheats. The conduction heat fluxes vary gradually with the macrolayer thickness for smaller superheats ($\Delta T = 20$ K) and the gradient of the conduction heat flux is small. The heat flux gradient is very high and varies drastically with the macrolayer thickness for large superheats ($\Delta T = 40$ K). The less heat energy is transported by the macrolayer of smaller values of superheats.

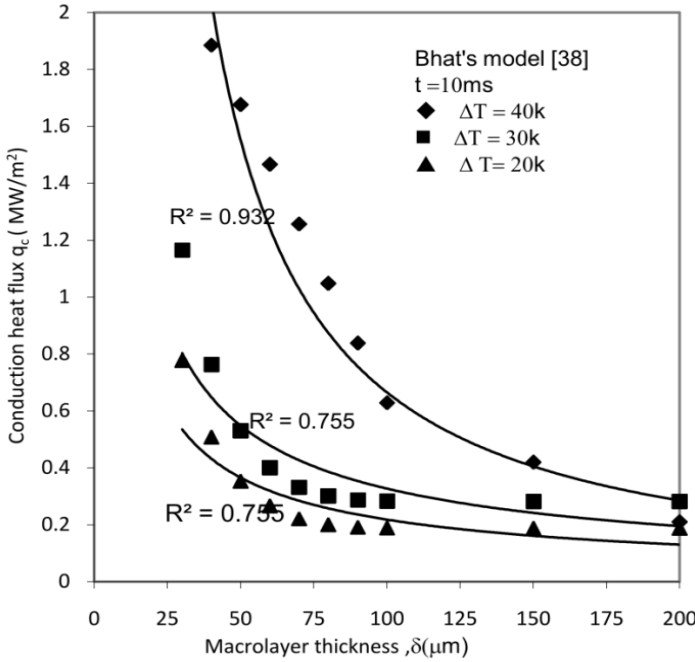

**Figure 3.** Effect of the macrolayer thickness on the heat flux for different superheats.

Figure 4 (present model) depicts the dependence of the heat flux on the macrolayer thickness at different superheats. The results were predicted for the wall superheats corresponding to 20 K, 30 K, and 40 K at a time of 10 ms. The heat flux of 1.6339 MW/m² was predicted for a wall superheat of 40 K and reduced to 0.4038 MW/m² with an increased macrolayer thickness to 200 μm. It is observed that a thinner macrolayer contributes to the constantly increased heat transfer rate. The heat transfer rate increases with the reduced macrolayer thickness. It is also perceived that the thicker macrolayer attributed to a small gradient of heat transfer and the curves turn out to be nearly straight for thicker macrolayers.

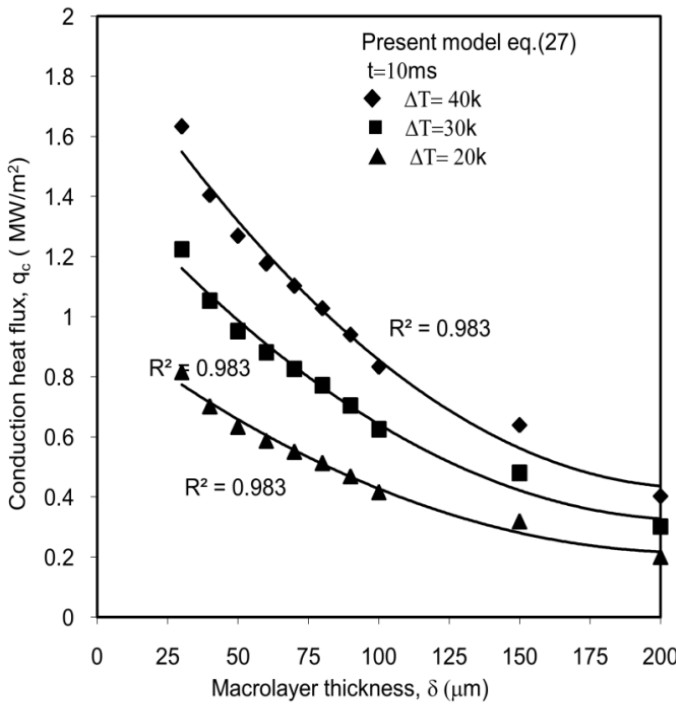

**Figure 4.** Effect of the macrolayer thickness on the heat flux for different superheats.

Figure 5 [38] depicts the transient conduction heat fluxes for macrolayer thicknesses of 50 μm, 75 μm, and 200 μm. The data were predicted at 40 K wall superheat. At a time of 40 ms, a macrolayer of thickness 50 μm predicts a flux of 0.5471 MW/m², whereas a thicker macrolayer ($\approx$75 μm) contributed to a heat flux of 0.3647 MW/m². A flux of 0.1883 MW/m² was observed for a thickness of a macrolayer equal to 200 μm at the same time of 40 ms. For shorter times at the equal wall superheats, the heat exchange rate is high. It decreases drastically as time increases. For longer times, the heat transfer rate becomes almost constant, with the curve approximately parallel to the time axis. For time approaching the total cycle period, the macrolayer transfers less heat but remains a strong function of the thickness and wall superheat.

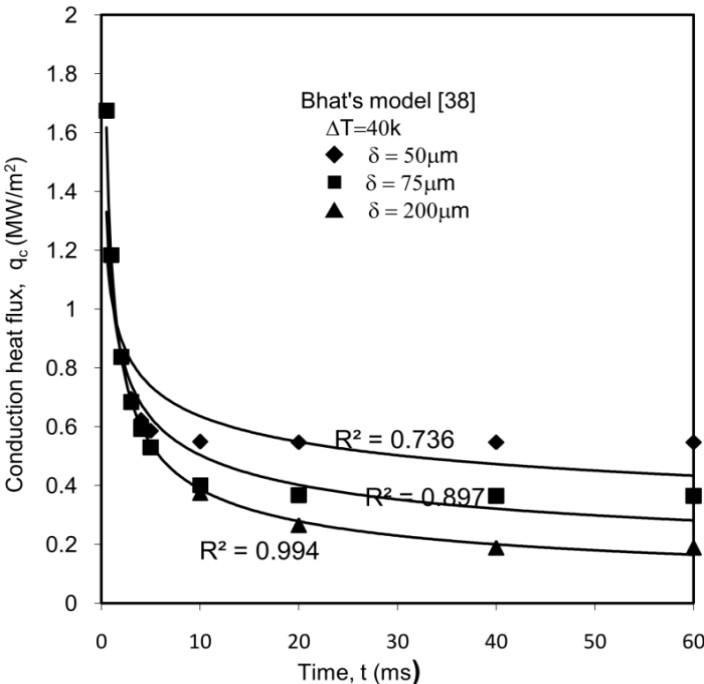

**Figure 5.** Transient conduction heat flux for different macrolayer thicknesses.

Figure 6 (present model) depicts the dependence of the fluxes on time for distinct values of macrolayer thickness. The data were predicted for macrolayer thicknesses of 50 μm, 70 μm, and 200 μm under a constant condition of superheat equal to 40 K. In general, a thinner macrolayer contributes to constantly enhanced heat transfer rates. In Figure 6, it seems that a thicker macrolayer imparts higher conduction heat flux for the initial and a very short period. The predicted flux gradient for a small period is tremendously very high and reduces significantly with time. The curves demonstrate that the initial heat transfer rates diminish considerably with time. The transient conduction heat fluxes of 1.9733 MW/m² and 1.1456 MW/m² were obtained for macrolayer thicknesses of 50 μm and 75 μm, respectively at constant t = 3 ms. However, the thickest macrolayer (200 μm) contributed to a conduction heat flux of 0.8375 MW/m² at the same time of 3 ms. As time approaches the total period (cycle), the rate curve becomes approximately parallel to the x-axis, signifying the reduced flux gradient with time.

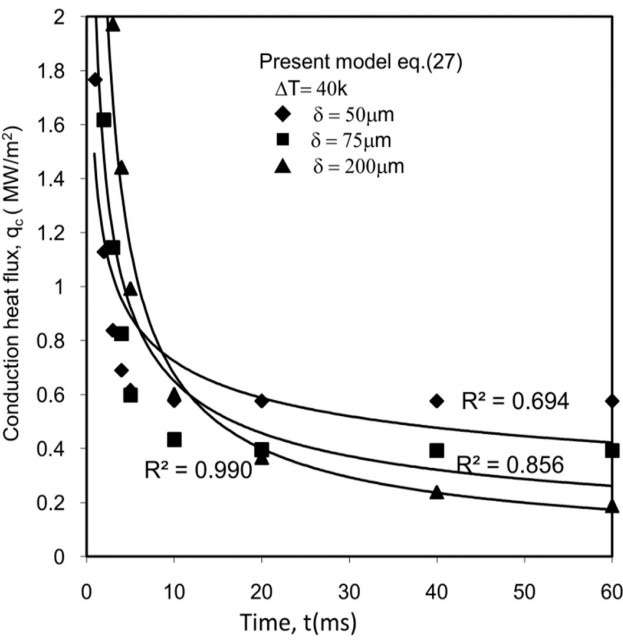

**Figure 6.** Transient conduction heat fluxes for different macrolayer thicknesses.

### 4.2. Average Heat Transfer Rates

The effect of superheat on the conduction heat flux (average) is depicted in Figure 7 [38]. The data were obtained for the total cycle time of 40 ms and with three macrolayer thicknesses of 30 μm, 40 μm and 50 μm. The average conduction heat flux increases with the raised wall superheat, and a thinner macrolayer leads to an increased heat transfer. Bhat's model predicted a maximal conduction flux (average) of 0.9218 MW/m$^2$ with a superheat of ~50 K at δ = 30 μm. A predicted heat flux of 0.8113 MW/m$^2$ was achieved at a relatively thicker macrolayer of 40 μm. From the analysis of the curves, it was established that the heat rate strongly depends on superheat. For the same cycle period and the same superheat, a thinner macrolayer transports less heat energy.

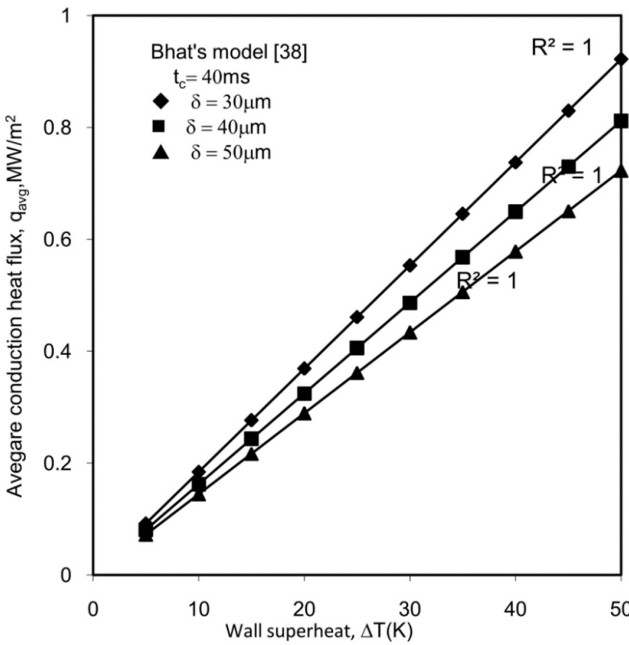

**Figure 7.** Effect of superheat on the average conduction heat flux at different δ with $t_c$ = 40 ms.

Figure 8 depicts the dependence of the heat flux (average) on superheat at various values of macrolayer thickness. The data were obtained for a departure time t = 40 ms with macrolayer thickness values of 50 μm, 75 μm, and 200 μm. The maximal average conduction heat flux of ~0.7212 MW/m$^2$ was achieved at a superheat =50 K at δ = 50 μm. The average conduction heat flux declined from 0.4448 MW/m$^2$ to 0.1927 MW/m$^2$ with increased macrolayer thickness from 75 μm to 200 μm at a constant ΔT = 45 K. Regardless of other parameters, the average conduction heat flux remains a strong function of the wall superheat. For macrolayers of different thicknesses, the $q_{avg}$ is less evident with small values of the wall superheat than with large values of the wall superheat.

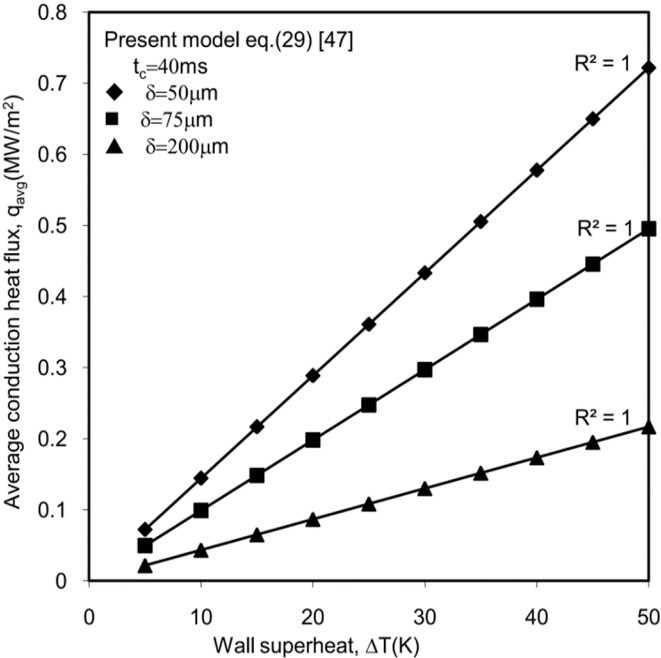

**Figure 8.** Effect of superheat on the average conduction heat flux at different δ with $t_c$ = 40 ms.

Figure 9 [38] depicts that the rate of heat exchange relies strongly on macrolayer thickness. The data were obtained for a total period of 40 ms with superheats of 20 K, 30 K, and 40 K. The flux gradient is larger for small values of the total cycle period as compared with large values of the total cycle period. The heat flux of 1.0457 MW/m$^2$ was predicted at a macrolayer thickness of 5 μm corresponding to 40 K wallsuperheat. The $q_{avg}$ declines from 0.7843 MW/m$^2$ to 0.4337 MW/m$^2$ with increased macrolayer thickness from 5 μm to 50 μm at $t_c$ = 40 ms.

Figure 10 depicts that the dependence of heat fluxes on the thickness of the macrolayer for the present model with wall superheats of ΔT = 20 K, 30 K and 40 K. It is demonstrated undoubtedly that the average heat flux (average) varies considerably with the thickness of the macrolayer. The gradient of the average conduction heat flux with macrolayer thickness for the present model is very high compared with the one observed for Bhat's model [38]. The heat flux reduces from 0.9411 MW/m$^2$ to 0.1712 MW/m$^2$ with raised δ from 30 μm to 200 μm, respectively. At any constant value of macrolayer thickness—say, δ = 50 μm—the predicted average conduction heat flux increases with the raised superheat. It is apparent that the heat rate reduces with increased δ.

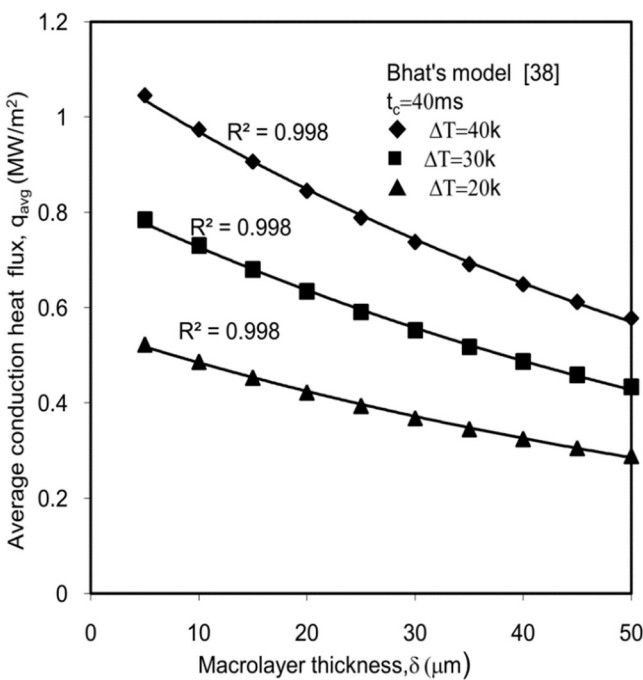

**Figure 9.** Effect of macrolayer thickness on the average conduction heat flux for different superheats with $t_c$ = 40 ms.

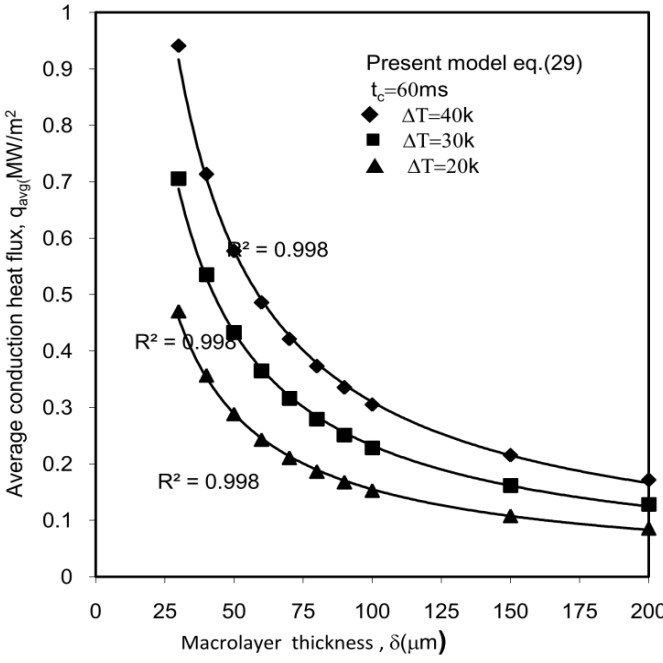

**Figure 10.** Effect of the macrolayer thickness on the average conduction heat flux for different superheats with $t_c$ = 60 ms.

### 4.3. Comparative Analysis of Predicted Heat Fluxes

In Figure 11, the results predicted by Bhat's constant macrolayer model [38] have been compared with the predicted values of the present model. The influence of the wall superheat on instantaneous conduction heat flux has been analyzed. In all cases of flux predicted by Bhat's model and the present model, the conduction flux remains a strong function of the wall superheat. A small deviation occurs between the flux predicted by the present model and Bhat's constant macrolayer model. The present

model for t = 1 ms with a macrolayer thickness of 50 μm and 150 μm predicts better results in terms of conduction heat flux compared with Bhat's model [38] for t = 5 ms corresponding to the same macrolayer thicknesses of 50 μm and 150 μm. The present model predicts a maximal flux of 1.5167 MW/m$^2$ compared with a lower heat flux of 1.3601 MW/m$^2$ at a ΔT = 50 K.

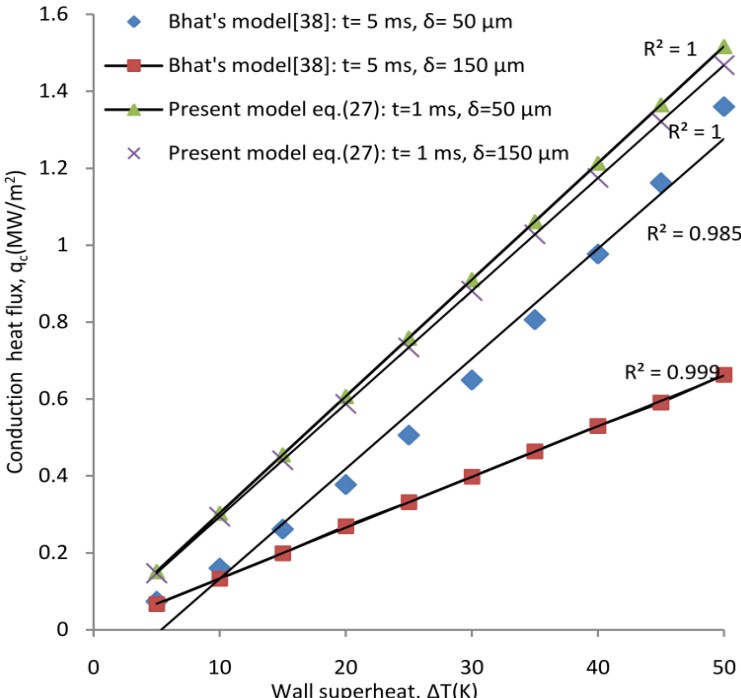

**Figure 11.** Relative dependence of the conduction heat flux on the wall superheat under constant conditions of time and macrolayer thickness.

Figure 12 depicts the comparative predicted values of Bhat's model [38] and that of the present model. The data were obtained for macrolayer thicknesses of 50 μm and 150 μm corresponding to a departure time of 10 ms. A significant difference was noticed between the results predicted by the Bhat model [38] and the present model for a thinner macrolayer of 50 μm. There is close agreement between the results predicted by both models for the macrolayers corresponding to a macrolayer thickness of 150 μm. The present model predicted a conduction heat flux of 0.4563 MW/m$^2$ compared with the nearly the same heat flux of 0.4683 MW/m$^2$ under the identical conditions of departure time (t = 10 ms) and the wall superheat (ΔT = 50 K).

Figure 13 illustrates the predicted findings of the present model and Bhat's model to analyze the dependence of the heat flux on the wall superheat. The data were obtained for wall superheats equal to 40 K and 30 K at a time of 10 ms. In all cases, increased macrolayer thickness contributed negatively to the conduction heat flux. Significant deviations in the heat fluxes given by the present model and the Bhat's model [38] were observed at ΔT = 40 K with a δ = 30 μm. The present model predicts better results compared with those predicted by Bhat's model [38] at ΔT = 30 K with $t_c$ = 10 ms. The heat flux of 1.2254 MW/m$^2$ was predicted for the present model compared with a lower value (1.1652 MW/m$^2$) for Bhat's model at ΔT = 30 K.

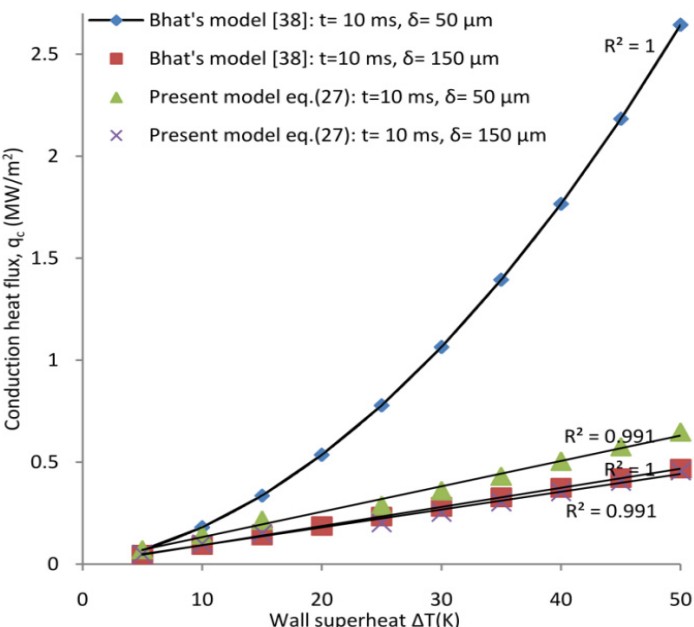

**Figure 12.** Relativedependence of the conduction heat flux on the wall superheat under constant conditions of time and macrolayer thickness.

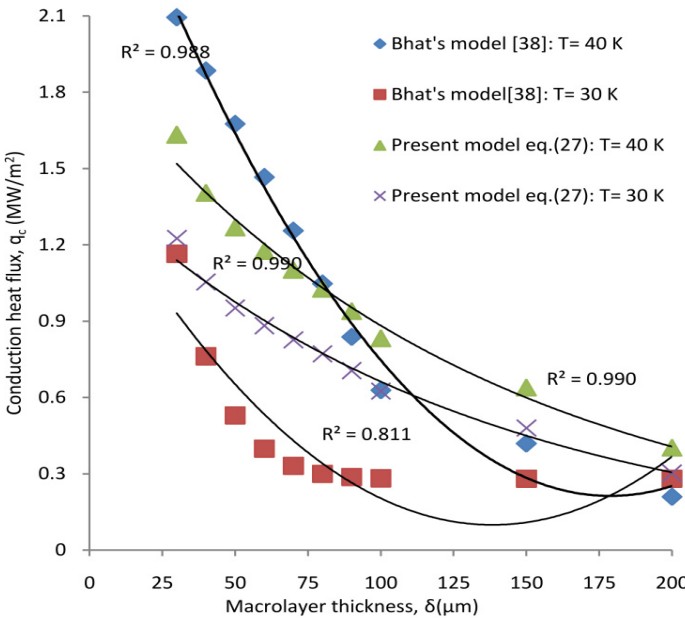

**Figure 13.** Relative dependence of the conduction heat fluxes on the macrolayer thickness at different superheats (t = 10 ms).

In Figure 14, the data were obtained for conduction heat fluxes predicted by the present model and by Bhat's model [38] for $\Delta T = 40$ K and 30 K at a t = 15 ms. The curves thus obtained signify that the current analytical model predicts the heat transfer at the heat flux corresponding to the nucleate boiling conditions. The current model predicts a heat flux of 0.9574 MW/m² compared with a lower value of 0.8755 MW/m² for Bhat's model at $\Delta T = 40$ K corresponding to $\delta = 50$ μm. On the other hand, the flux of 0.7181 MW/m² was obtained using the present model compared with a flux 0.6559 MW/m² obtained for Bhat's model at $\Delta T = 30$ K. The predicated heat fluxes are better for a thinner macrolayer as compared with the values corresponding to a thicker macrolayer.

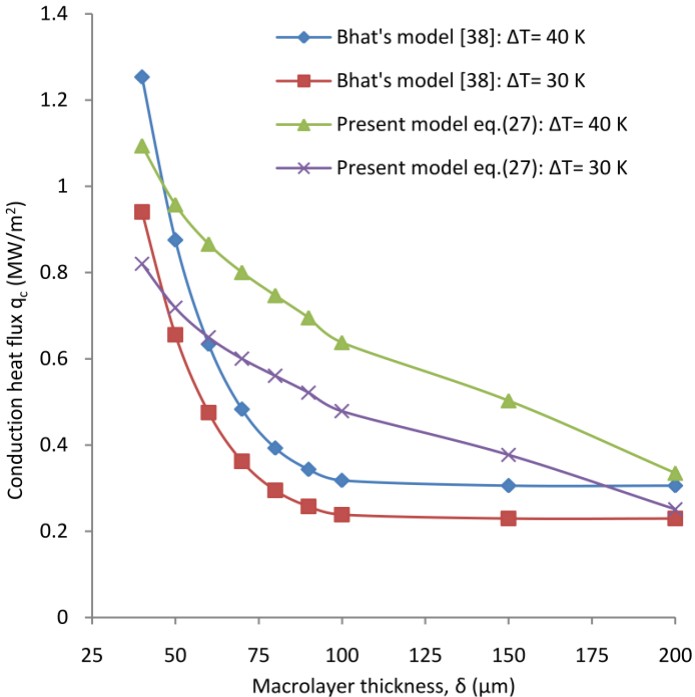

**Figure 14.** Relative dependence of the conduction heat fluxes on the macrolayer thickness at different superheats (t = 15 ms).

## 5. Conclusions

The present analytical model provides a good prediction of the heat transfer at high heat fluxes. The conduction heat rate, as predicted by the present model and by Bhat's model, is relying strongly on wall superheats. That is, increased superheats demonstrate constantly higher rates of heat exchange, but the effect tends to decline as the departure time is reached. A thinner macrolayer transports more heat energy as compared with a thicker macrolayer under the similar conditions of ΔT. The influence of δ on the conduction heat flux predicted by Bhat's model is less pronounced for a thicker macrolayer (150 μm to 200 μm) compared with that pronounced by the current model. For a smaller t, the heat transfer rate is high, and it reduced drastically with time as predicted by both the present model and Bhat's model at the same ΔT.

It has been suggested that the heat flux (average) increases with raised ΔT and a thinner macrolayer contributed significantly to the heat transfer. The average flux predicted by the present model is in close agreement with that obtained using Bhat's model at δ = 50 μm for ΔT = 50 K. The present model at t = 1 ms predicts better results in terms of the conduction heat flux compared with Bhat's model [38] at t =5 ms corresponding to the same macrolayer thicknesses of 50 μm and 150 μm. It is apparent that the average heat rate reduces with δ, indicating that a thinner macrolayer contributes to increased heat fluxes. The present model predicts a heat flux of 0.9574 MW/m$^2$ compared with a lower value ~0.8755 MW/m$^2$ predicted by Bhat's model for ΔT = 40 K at δ = 50 μm. For a higher ΔT, which corresponds to a nucleate boiling state, the outcomes of the present model closely agree with the findings of Bhat's model.

**Author Contributions:** M.D. and M.K.A.M. have contributed equally to the research articles.

**Funding:** This research received no external funding.

**Acknowledgments:** Authors are grateful to the Deanship of Scientific Research, King Khalid University, Abha, Kingdom of Saudi Arabia.

**Conflicts of Interest:** There is no conflict of interest to declare.

## Nomenclature

| | |
|---|---|
| $h_{fg}$ | Latent heat of vaporization, J/kg |
| k | Thermal conductivity, W/mK |
| q | Heat flux, MW/m$^2$ |
| $q_c$ | Conduction Heat flux, MW/m$^2$ |
| $q_w$ | Wall Heat flux, MW/m$^2$ |
| $q_{avg}$ | Average Conduction Heat flux, MW/m$^2$ |
| $R^2$ | Determination coefficient |
| t | Time, ms |
| $t_c$ | Total cycle time, ms |
| $\Delta T$ | Wall superheat, K |
| T | Temperature, K |
| $T_s$ | Saturation temperature, K |
| $T_w$ | Wall temperature, K |
| V | vapor liquid interface velocity, m/s |
| *Greek Symbol* | |
| $\alpha$ | Thermal diffusivity of liquid, m$^2$/s |
| $\delta$, $\delta o$ | Macrolayer thickness, initial value, μm |
| θ | Dimensionless temperature |
| *Subscripts* | |
| avg | Average |
| c | Conduction |
| i | Input |
| s | Saturation |
| w | wall |

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
