# Peer review of "Developing a Mathematical Model for Nucleate Boiling Regime at High Heat Flux"

_processes, doi:10.3390/pr7100726_

Round 1

Reviewer 1 Report

All the figures and graphs need to be more descriptive as to what is contained in the figures and the graphs. This is to make sure that the readers can get a general idea of what is going on once they read the caption and they can reference to the manuscript for more information.  The introductioncan be made better by elaborating on the literature review. The authors just wrote one sentence on previous research done, which only states the title of the research done, instead of what was done and what was found in previous studies. The purpose of the study in lines 101-105 should be elaborated. The authors should mention why the development of this model is important and how it fills in the gap between previous studies in the field. The authors should mention and reference the origins of equation (26). In Figure 5, there are only two curves which correspond to 50 microns and 150 microns. However, 200 microns is stated in the legend. This reviewer understands that this figure is referenced from the Bhat model and not the work of the authors, but it is the authors’ responsibility to explain this discrepancy in the text or the caption of the figure. Both in the conclusion and the abstract, the authors mention the comparison of between the present model and Bhat model, but the significance to the nucleate boiling regime at high heat flux is not shown clearly in the manuscript. In Figure 10, it seems that the conduction heat flux of the thicker macrolayer is higher than that of the thinner macrolayers at a low time. However, the opposite case seems to be imminent for higher time. The authors need to explain the reason behind this scenario in the discussion. For the result and discussion section, the authors are mainly comparing their model to the Bhat model. If this is the case, then consistency is required for comparison. This is lacking in lines 199 and 206, where comparison was made for different times. This is unacceptable because it leads to inconsistency of data. The authors should compare the Bhat model to the present model at 10ms first, and then elaborate on the predictions of the models at t=40ms if necessary. This comparison inconsistency also happens at Figure 9 and 10, where in Figure 9, the thicknesses are 50 microns, 75 microns and 200 microns while in Figure 10, the thicknesses are 50 microns, 75 microns and 200 microns. It is regrettable that many grammatical and spelling errors were found. Please fix them to be at an academic standard. For the effect of microlayer thickness on average conduction heat flux in Figure 14, the results for present model varies a lot from Bhat model. In the corresponding discussion paragraph (lines 250-255), the authors only mention the decrease of average conduction heat flux with increase of macrolayer thickness, but not the pattern of variation. The authors should elaborate why the curves in Figure 14 take a quadratic shape when the one from Bhat model seem linear.

Author Response

The point-by-point response to the reviewer's comment has been provided.Please see the attachment.

Reviewer 2 Report

When summarizing from equation 15 to equation 16, the term Sr is removed. The reason is that there is no heat source from outside. Explain the physical meaning of Sr. The difference between the model in Reference 38 and the current model is needed first. For example, please explain the physical properties of equations 2 and 27. Explain the results and effects of the conduction heat flux that you actually want to compare. In equation 16, the x and z terms can be ignored because there is no temperature change in the x and z directions. But there is an error in equation 16. Check and correct Equation 16 Figures 5 and 6 show the results of comparing Bhat's model with the current model. Figure 5 shows the results of conduction heat flux with wall temperature for 5ms. However, Figure 6 is not considered to be the same condition as Figure 5. If you have a different time reason, explain the physical reason for it. Or perform a comparison under the same conditions. I think that the result of analysis under the same conditions is very difficult to analyze with other graphs. The results of Figure 5 ~ Figure 10 on one graph and reference (38) and the current model on one graph are more relevant. Check the legend in Figure 15, 16. The symbol and symbol of the graph are somewhat unclear.

Author Response

The point-by-point response to reviewer's comments has provided.Please see the attachment.

Round 2

Reviewer 1 Report

What does CHE in lines 55 and 74 mean? In the sentence “The predicted results agree very well with experimental findings in CFD study of cryogenic fluid by Xiaobin and others [22].”, what do the authors mean by the predicted results? If it refers to the previous cited study? Can the authors explain the reason behind this sentence “The dependence of macrolayer thickness on conduction flux is less evident for smaller wall superheats.” in line 276?

Author Response

Dear Editor,

Point-by-point response to the reviewer's comments has been provided.Please see the attachment.

regards,
